# Nano- and Micro-Encapsulation of Long-Chain-Fatty-Acid-Rich Melon Seed Oil and Its Release Attributes under In Vitro Digestion Model

**DOI:** 10.3390/foods12122371

**Published:** 2023-06-14

**Authors:** Asliye Karaaslan

**Affiliations:** Vocational School of Organized Industrial Zone, Food Processing Programme, Harran University, 63300 Sanliurfa, Turkey; asliyegumus@harran.edu.tr

**Keywords:** nano-liposome, *Cucumis melo*, long-chain fatty acids, in vitro release, encapsulation

## Abstract

Melon seed oil (MSO) possesses plenty of long-chain fatty acids (LFCAs, oleic–linoleic acid 90%), remarkable antioxidant activity (DPPH (0.37 ± 0.40 µmol TE/g), ABTS (4.98 ± 0.18 µmol TE/g), FRAP (0.99 ± 0.02 µmol TE/g), and CUPRAC (4.94 ± 0.11 µmol TE/g)), and phenolic content (70.14 ± 0.53 mg GAE/100 g). Encapsulation is a sound technology to provide thermal stability and controlled release attributes to functional compounds such as plant seed oil. Nano-sized and micro-sized capsules harboring MSO were generated by utilizing thin film dispersion, spray drying, and lyophilization strategies. Fourier infrared transform analysis (FTIR), scanning electron microscopy (SEM), and particle size analyses were used for the authentication and morphological characterization of the samples. Spray drying and lyophilization effectuated the formation of microscale capsules (2660 ± 14 nm, 3140 ± 12 nm, respectively), while liposomal encapsulation brought about the development of nano-capsules (282.30 ± 2.35 nm). Nano-liposomal systems displayed significant thermal stability compared to microcapsules. According to in vitro release studies, microcapsules started to release MSO in simulated salivary fluid (SSF) and this continued in gastric (SGF) and intestinal (SIF) environments. There was no oil release for nano-liposomes in SSF, while limited release was observed in SGF and the highest release was observed in SIF. The results showed that nano-liposomal systems featured MSO thermal stability and controlled the release attributes in the gastrointestinal system (GIS) tract.

## 1. Introduction

Fruits constitute a significant part of the human diet as they contain plenty of health-promoting compounds such as vitamins, minerals, antioxidative compounds, amino acids, and essential fatty acids, besides their desirable taste, aroma, and appearance. A big portion of fruit harvest is transported to factories to manufacture a myriad of canned, dehydrated, and frozen products. A large amount of by-products are produced via the industrial processing of fruit tissues. By-products or side streams cause negative environmental and economic impacts if they are not evaluated or managed properly. Modern food processing perspectives and the sustainable bioeconomy involve the valorization of by-products, and side streams emerge during the production of packaged food commodities. There has been an exponential increase in plant-based food processing industry by-products, which should be evaluated properly to overcome environmental issues and to generate novel sources for the production of functional foods or supplements. Evaluating food processing by-products and integrating them into value-added products could serve as the development of the circular economy, which has the potential to create economic growth.

The sugar/acid ratio, color, taste, and widely distributed plantation make certain fruit types worthy for industrial processing. Sour cherry, pomegranate, apple, peach, grape, apricot, and melon (*Cucumis melo* L.) are utilized as raw materials in the production of many packaged food commodities. *Cucumis melo* L. originates in Asia [1] and belongs to the *Cucurbitacae* family, and China, the USA, and Türkiye are the largest melon producers. Melon mesocarp, pulp, and certain melon-based extracts can be evaluated in the manufacturing of jam, juice, and canned and dehydrated products [2]. The agri-food melon processing chain generates a large amount of food by-products. As melon is one of the most widely distributed and planted fruit types, great effort is required to develop innovative strategies for the utilization of melon processing by-products to acquire functional compounds and produce foods with high nutritional quality [3]. Environmental and economic concerns inspired the researchers to discover alternative approaches to valorize such by-products. Extracts and powders derived from side streams or by-products could be incorporated into highly demanded dietary supplements and functional foods. Peel (outer skin) and seed emerge during the processing of melon fruit. A detailed analysis of the valorization of melon by-products has been reviewed recently [2]. Melon seeds equal 10% of the total fruit weight and could be labeled as a rich source of biologically important components such as carotenoids and phenolic compounds. The highest total phenolic concentration was detected in the seeds [4] in comparison with the peel and the pulp. Melon seeds contain tocopherol fractions ranging from 2.23 to 6.88 for α-tocopherol and from 17.73 to 63.08 mg/100 g of oil for γ-tocopherol [2]. A high essential fatty acid content highlights the nutritional value of melon seeds and it should be noted that seeds contain the largest amount of long-chain fatty acids (LCFAs) when considering the whole fruit [5]. The remarkable vitamin, sterol, flavonoid, phenolic, and ω fatty acid contents of MSO comprise distinctive antimicrobial, free radical scavenging, and antihyperlipidemic potential [2,6]. The fatty acids found in MSO are of many kinds, but linoleic and oleic acids represent almost 85–90% of the total fatty acids existing in MSO [3,7,8,9]. The excessive LCFA composition of MSO indicates its desirable nutritional status. Therefore, the utilization of fruit processing by-products, for instance, melon seeds, could alleviate the industry’s demand for alternative sources to obtain functional vegetable oils [6].

However, MSO is prone to negative effects of heat, light, and oxygen due to the excessive existence of mono- and polyunsaturated fatty acid content. Proper storage conditions or conservation methods should be implemented to preserve its nutritional quality [10]. Encapsulation technology involves the entrapment of food value or biologically active sensitive core material within a stable polymeric network and also facilitates the homogenous incorporation of hydrophobic food macromolecules into the food matrix. It has commonly been recognized to improve the storage endurance of sensitive compounds [11]. Encapsulation can be utilized to enhance the stability of sensitive compounds such as ω fatty acids and antioxidative compounds, and it enables the incorporation of hydrophilic and hydrophobic components within the same food matrix. Encapsulation technology can be utilized to provide improved stability, protection against external conditions, endurance, and desirable sensorial properties of the core material [12]. Certain antioxidative compounds, vitamins, anthocyanins, and phenolics are digested within the digestion tract. In particular, high acidic environments result in the decomposition of biologically active compounds [13]. Therefore, the controlled release and targeted delivery of food value or bioactive compounds is required to protect such compounds. Lately, nano-liposomal systems have been adopted to provide better stability and release attributes for functional compounds. Nano-liposomes might be defined as lipid-based nano-carriers that can be used to encapsulate bioactive, functional, and food-value compounds with different chemistry [14]. Nano-liposomal systems provide an opportunity for the targeted delivery and controlled release of functional compounds, while protecting the nutritional value and quality of the oil [15]. While MSO is a rich source of health benefiting sensitive natural compounds such as long-chain fatty acids (LFCAs) and phenolic compounds, and also displays strong antioxidant activity, there is no study in the literature depicting the encapsulation and release characteristics of MSO. Spray drying, lyophilization, and liposomal encapsulation techniques were engaged to find capsules carrying MSO. The characteristic properties of the capsules containing MSO were measured in the context of the study and their release behavior was simulated within an in vitro GIS model.

## 2. Materials and Methods

### 2.1. Materials

Melon seeds were obtained from a local market in Şanlıurfa (Türkiye). The melon seeds were dehydrated at room temperature for 5 days and then kept in a refrigerator at 4 °C using an air-tight container prior to analyses. The analytical-grade chemicals were obtained from Sigma or Merck, unless otherwise stated. Soybean lecithin, gum arabic, and maltodextrin were purchased from Sigma Aldrich Co. (St. Louis, MO, USA)). 

### 2.2. Oil Extraction

#### 2.2.1. Soxhlet Extraction

The seeds (20 g) were powdered in a coffee grinder and mixed with n-hexane (200 mL). Extraction was conducted in a water bath at 80 °C. After 360 min, this mixture was filtered and the organic phase was removed at 50 °C using a rotary evaporator (Buchi, Flawil, Switzerland). The hexane residues in the obtained oils were evaporated at 50 °C in a vacuum drying oven (DZF-6020-T, Binglin Co., Ltd., Shanghai, China) [16].

#### 2.2.2. Microwave-Assisted Extraction

A glass microwave apparatus containing 20 g grinded seeds and 200 mL n-hexane was placed in the device (Sineo, Mass II Plus, Shanghai, China) with temperature control. The oil extraction conditions were 500 watt, 80 °C, and 30 min for power, temperature, and time, respectively. The hexane phase was evaporated at 50 °C using a rotary evaporator and then the extracted oil was exposed at 50 °C in a vacuum drying oven for removing the hexane residues.

#### 2.2.3. Ultrasound-Assisted Extraction

The ultrasound-assisted extraction was conducted in a laboratory-type ultrasonic bath (Wiseclean WUC-D10H, Wertheim, Germany). For this, a glass bottle in the presence of 20 g grinded seeds and 200 mL n-hexane was placed in the ultrasonic bath. The extraction temperature (80 °C) and extraction time (30 min) were fixed during this process. Hexane evaporation was conducted at 50 °C using a rotary evaporator and a vacuum drying oven, respectively.

#### 2.2.4. Combination of Microwave- and Ultrasound-Assisted Extraction

Firstly, microwave-assisted extraction was applied for 15 min and the remaining extraction time (15 min) was continued in the ultrasonic bath. All conditions for both systems except for time were fixed, as mentioned before.

### 2.3. Oil Yield

The oil yield was calculated according to the method of Tian et al. [17] using the following equation.
$$Oil\;yield\;\left(\%\right)=\frac{extracted\;oil}{initial\;seed\;mass}\times 100$$

### 2.4. Oxidation Parameters, Color, and Bioactive Properties of MSO

The NP EN ISO 3960:2004 and NP-1819:1984 Portuguese standards were used for the calculation of the PV and p-AV values and the TOTOX value was determined by following the equation TOTOX = 2 × PV + p-AV. The K232 and K270 values were detected using the spectrophotometric method. For this, 25 mL isooctane was transferred to a glass tube containing 0.25 g oil. After the oil was completely dissolved, the absorbance of this mixture was read at 232 and 270 nm by utilizing a spectrophotometer (Model UV-1700, Shimadzu Corp., Kyoto, Japan) [18]. The values of peroxide and p-anisidine were calculated as described previously [18]. The color of the MSO samples was determined in terms of the CIE L*, a*, and b* values by utilizing a colorimeter (Color Quest, Hunter Associates Laboratory, Inc., Reston, VA, USA) [19]. The total phenolic content of the samples was measured by employing a spectrophotometric method using Folin–Ciocalteu reagent, and a gallic acid curve was used as a reference standard to calculate the amount of phenolics [20]. Briefly, 3.9 mL of DPPH methanolic solution (2.5 mg/100 mL) was included into a volumetric flask containing 0.1 mL diluted sample or Trolox to measure the radical scavenging ability of the MSO. The absorbance was measured at 515 nm after 30 min of incubation time through the use of a spectrophotometer (Model UV-1700, Shimadzu Corp., Kyoto, Japan). The obtained results were given as µmol TE/g. ABTS agent (30 mg) was dissolved in potassium persulfate (2.46 mM) at a volume of 7.8 mL to prepare the ABTS solution. Diluted samples (1:20) were mixed with ABTS solution (1950 μL) and incubated for 6 min prior to measuring absorbance at 734 nm (Model UV-1700, Shimadzu Corp., Kyoto, Japan). The obtained results were given in µmol TE/g [21]. A spectrophotometric assay was employed to carry out the CUPRAC assay. Diluted extracts (0.4 mL; 0.05–2.5 g/mL), copper(II) chloride (1 mL, 0.01 M), ethanolic neocuproine (1 mL 7.5 × 10^−3^), and ammonium acetate (1 mL, 1 M, pH 7) were combined within a volumetric flask. The prepared solution was incubated for 30 min at room temperature after adding 0.7 mL of distilled water. A UV-vis spectrophotometer was used to measure the absorbance at 450 nm (Model UV-1700, Shimadzu Corp., Kyoto, Japan) [22]. A total of 25 mL of acetate solution (30 mM), 2.5 mL of 2,4,6-Tris(2-pyridyl)-s-triazine (10 mM), and 2.5 mL of iron(II) chloride (20 mM) were mixed to form the FRAP buffer. A total of 150 μL of the diluted sample or Trolox were incorporated into 2850 μL of FRAP buffer and left to incubate for 30 min at room temperature. A UV-vis spectrophotometer (Model UV-1700, Shimadzu Corp., Kyoto, Japan) was utilized to measure the absorbance at 593 nm [23].

### 2.5. Fatty Acid Composition

Sodium methoxide was used for the preparation of methyl esters of the fatty acids. The fatty acid methyl ester of seed oils was injected into GC-FID (Shimadzu, GCMSQP2010). The fatty acid profile was detected using a fused-silica capillary column (30 m 0.25 mm) coated with 0.25 mm dimethylpolysiloxane (RTX^®^-5MS, Restek, Bellefonte, PA, USA). The column conditions were 40 °C (kept for 2 min), and shifted to 240 °C in 3 °C/min increments. The other parameters, namely injector (250 °C), interface (250 °C), and detector (220 °C) temperatures, were fixed. Helium was the carrier gas (1.0 mL/min) and the split ratio was 1/20 [24].

### 2.6. Generation of Nano-Liposomal Systems Containing MSO

The thin film dispersion method was employed to manufacture the nano-liposomal systems, as previously reported by Nahr et al. [25], with minor modifications. Soybean lecithin and MSO were dispersed in chloroform and incubated overnight at 4 °C to generate the complex structure. After the incubation time, chloroform was removed by utilizing a rotary evaporator (Buchi, Flawil, Switzerland) at 40 °C until a visible thin film structure was cast. Sodium phosphate buffer (10 mM, pH 7) was used to complete the film hydration step. The thin films were then subjected to a Sonopuls UW2070 ultrasonicator equipped with MS72 probes (Bandelin electronic GmbH & Co. KG; Berlin, German). The Lecithin/MSO ratio ranged from 1:1 to 1:16 (1:1, 1:4, 1:8, 1:16) and the sonication time ranged from 1 min to 12 min (1, 3, 6, 9, 12). The encapsulation efficiencies were used as the dependent variables to find the best conditions for the production of nano-liposomal systems. Encapsulation efficiency for nano-liposomal systems was measured according to the method reported by Choudrary et al. [15].

### 2.7. Encapsulation of MSO via Spray Drying and Lyophilization

For the production of MSO-containing spray-dried and lyophilized capsules, a method that was previously developed in our laboratory was utilized [26]. A composition of gum arabic (GA)/maltodextrin (MD) (1:11), 1 to 5 core to wall material ratio, and an inlet temperature of 195 °C were used for the production of spray-dried capsules. Lyophilized samples were produced using the same GA:MD composition and core to wall material ratio by employing a freeze dryer operating at −35 °C under −70 pa, and the resulting lyophilized samples were grounded into powder prior to analyses. The encapsulation efficiency was calculated according to the method described by Başyiğit et al. [26].

### 2.8. FTIR Spectroscopy

FTIR spectroscopy (IRTracer-100, Shimadzu Co., Kyoto, Japan) analyses were carried out to identify and characterize the specific bonds and groups present in lecithin, maltodextrin, gum arabic, and MSO, in addition to lyophilized, spray-dried, and nano-liposomal samples. The FTIR spectra were recorded between 600 and 4000 cm^−1^ with a resolution of 1 cm^−1^ at room temperature [27].

### 2.9. Particle Size, Zeta Potential, and Polydispersity Index Analyses

A zetasizer (Nano ZS90, Malvern Instruments, Malvern, UK) was operated to find out the particle size, zeta potential, and polydispersity index of nano-liposomal systems [28]. The particle size distributions of lyophilized and spray-dried capsules were measured by employing a mastersizer (Mastersizer 2000, Malvern Instruments Co., Malvern, UK).

### 2.10. Scanning Electron Microscopy

Morphological characterizations of the samples were carried out by utilizing scanning electron microscopy (SEM) (ZEISS Sigma 300 Field Emission SEM, Oberkochen, Germany). Palladium-plated samples were deposited in the SEM operated under a high vacuum. The structures of the samples were revealed at 1.00 kx magnification [29]. 

### 2.11. In Vitro Gastrointestinal Release

An in vitro GIS tract model system was designed to visualize the release behavior of MSO during oral, gastric, and intestinal digestion. The simulated fluids (saliva fluid (SSF), gastric fluid (SGF), and intestinal fluid (SIF)) were assembled as described by Minekus et al. [30]. To prepare SSF, 3 g nano-liposomal system was mixed in 4.8 mL simulated saliva electrolyte solution. Afterward, α-amylase (enzyme activity: 75 U/mL) was included into the solution to mimic the enzymatic activity within the oral phase in addition to 25 µL calcium chloride (0.3 M). The pH of the solution was brought to 7.0 by including an appropriate amount of NaOH (1 M) and the total volume was adjusted to 9 mL with sterile distilled water. The final solution was maintained at 200 rpm at 37 °C in a water bath for 2 min.

To establish the gastric phase, 4.5 mL of saliva-digested nano-liposomal systems (oral bolus) was added to 4.095 mL simulated gastric electrolyte solution. The final SGF was created by including pepsin (enzyme activity: 2000 U/mL), calcium chloride (2.25 µL, 0.3 M), and distilled water (402.75 µL) into the mixture. The pH of the mixture was adjusted to 3.0 with hydrochloric acid (1 M). The gastric phase was mimicked by incubating the prepared mixture at 200 rpm for 2 h at 37 °C in a water bath.

The digested sample in the gastric phase (kimus, 4.5 mL) was mixed with simulated intestinal electrolyte solution (4.1625 mL) to prepare the intestinal phase. The final mixture was created by adding enzymes (trypsin, lipase, and α-amylase with 100 U/mL, 2000, and 2000 U/mL activities, respectively), bile salts (45 mg), calcium chloride (10 µL, 0.3 M), and distilled sterile water (328.50 µL). The pH of the mixture was set to 7.0 through the use of NaOH (1 M). The digestion phase was carried out in a water bath set to 37 °C at 200 rpm for 2 h.

At the end of each phase, digestive fluids were centrifuged at 5000× *g* rpm for 15 min. The amount of MSO in each phase was measured to determine the gastrointestinal release behavior of the capsules. The MSO released in the digestive phases was measured at 270 nm and the oil content was calculated according to a previous study [15]. The ratio of the released MSO into the simulated tract was measured using the below equation.
$$Release\;ratio\;\left(\%\right)=\frac{released\;oil\;after\;digestion}{initialoil}\times 100$$

The same procedure was applied to reveal the release characteristics of the spray-dried and lyophilized samples containing MSO within the simulated digestive tract.

### 2.12. Statistical Analysis

All experiments and analyses were conducted at least in triplicate and repeated twice. The differences between the means were determined using one-way analysis of variance (ANOVA) and Duncan’s multiple comparisons test at the confidence level of 95% (*p* ≤ 0.05). The obtained data were analyzed using a statistical package for Windows (SPSS Inc., version 22.0, Chicago, IL, USA). 

## 3. Results and Discussion

### 3.1. Extraction Efficiency, Color Characteristics, and Bioactive Content of Melon Seed Oil

Novel extraction techniques were adapted to the bio-separation processes to dissociate food value fractions from plant or animal tissues. Microwave ultrasound applications or their combinations are common methods employed for the extraction of seed oils [31]. Therefore, a similar approach was embraced in this study to determine an efficient method to extract oil from melon seeds. Soxhlet extraction (SE), microwave-assisted extraction (MWAE), ultrasound-assisted extraction (USAE), and a combination of microwave and ultrasound-assisted extraction (UMAE) were utilized for the removal of oil from melon seeds. The MSO yield ranged from 18.75 to 22.85%. The oil extraction efficiency from melon seeds was around 30%, as reported by Da Silva and Jorge [32]. The obtained oil extraction efficiencies were low to a limited extent when compared with the previous results [2,32]. The observed differences between extraction efficiencies could be attributed to the varietal or seasonal variations. USAE resulted in the isolation of the highest amount of oil from the melon seeds among the three employed methods. The USAE method ensured a high yield and simultaneously a high quality of the final product, such as low peroxide value (3.96 ± 0.03 meq/kg oil), p-anisidine (2.19 ± 0.12), TOTOX value (10.11 ± 0.22), K232 (0.45 ± 0.01), and K270 (0.11 ± 0.04) values (Table 1). The effect of extraction on the color parameters of the melon seed oil is given in Table 2. There was no obvious correlation between extraction method and color parameters of the extracted oil. Efficiency, yield, and bioactive properties of plant extracts are strongly dependent on the employed extraction method, since different extraction techniques have varied capabilities to release the natural compounds trapped within the plant tissues [33]. The phenolic content of plant extracts is one of the parameters indicating its functional potential. Reactive oxygen species and reactive nitrogen species are two groups of free radicals that might cause the formation of degenerative disorders within living organisms. It has been reported that phenolic compounds could protect the human body against the detrimental effects of free radicals on cells, genetic material, membranes, and tissues [34], as they display radical scavenging activity and reducing power. The correlation between high phenolic content and strong antioxidant capacity has also been reported in many studies [35,36]. Therefore, the phenolic contents of melon seed oils extracted using different methods were determined. It was found that both ultrasound and microwave treatments have a profound effect on the phenolic content of the extracted oils. The USAE brought about the highest total phenolic content (70.14 ± 0.53 (mg GAE/100 g)) in MSO compared to the samples extracted using the other methods (Table 3). The DPPH and ABTS methods were employed to evaluate the radical scavenging potential of the melon seed oil while the CUPRAC and FRAP methods were used to determine the reducing power of the oil samples. All four antioxidant capacity measurement techniques indicated that UASE oil has the highest antioxidant activity and potential to deactivate free radicals.

Ultrasound-assisted extraction is considered to be one of the technologies that has the capability to isolate oil from plant tissues with low energy consumption and improved oil extraction yield [37]. The cavitations generated via the ultrasound treatment led to the development of localized high temperature and pressure, which facilitate the release of intercellular material such as oil [38]. The obtained data in the current study showed that ultrasound-assisted extraction resulted in the highest oil yield, phenolic content, and antioxidant activity among the methods employed to extract MSO and could be used to obtain a high amount of MSO from the melon seeds. 

### 3.2. Fatty Acid Composition of MSO

Melon seeds could be considered to be a valuable by-product as they are a good source of essential fatty acids [2]. Physicochemical and nutritional attributes of oil are specified by the composition and sort of fatty acids bounded to the glycerol backbone present on triacylglycerol. Oleic and linoleic acids are classified as long-chain fatty acids (LCFAs), which could promote human health by controlling cholesterol metabolism which is associated with the occurrence of cardiovascular diseases [39]. Górnaś and Rudzińska [40] investigated the fatty acid composition of honeydew melons, which contain a high amount of linoleic acid (59.04%), followed by oleic acid (24.71%) and palmitic acid (9.57%). Mallek-Ayadi et al. [41] similarly concluded that melon seeds include linoleic acid (68.98%), oleic acid (15.84), and palmitic acid (8.71%). Therefore, GC-FID analyses were carried out to dissect the fatty acid profile of the MSO obtained in this study. Four different MSOs derived from soxhlet, USAE, UMAE, and USAE-UMAE combination were subjected to chromatographic screening to reveal the existing profile. The calculated oil extraction efficiency of melon seeds was around 22%. A chromatographic reading indicated that MSO contains a remarkable concentration of LFCAs. The major fatty acids found in MSO extracted with USAE were linoleic acid (73.56%), oleic acid (15.21%), and palmitic acid (9.95%) (Table 4). The total percentage of LFCAs existing in MSO was 89.77%. The high polyunsaturated acid and LCFA content highlight the biological importance of MSO, which could be used as a rich source of LCFAs to enhance the functional value of foods or dietary supplements. The results obtained in this study show that MSO is an exceptional natural source for LFCAs and thus could be incorporated into various encapsulation systems to serve as a functional core material.

### 3.3. Development of Nano-liposomal and Microcapsule Systems Containing LFCAs

An experimental design was conceived to generate lecithin-based nano-liposomal systems containing MSO (Table 5). The MSO/Lecithin ratio and sonication time were used as variables to fabricate nano-liposomal systems with high encapsulation efficiency (EE). The MSO/Lecithin ratio ranged from 1:1 to 1:16, while the sonication time varied from 1 min to 12 min. The maximum EE was achieved in the nano-liposomal systems constructed with a 1:8 MSO/lecithin ratio. In the second step, the effect of sonication time on the EE of the nano-liposomal system was investigated. Increasing the sonication time from 1 min to 9 min positively affects the EE of nano-liposomes. A further increase in sonication time to 12 min negatively influenced the EE, and thus 9 min of sonication time and a 1:8 wall material to core ratio were selected as the best conditions to manufacture nano-liposomal systems. It was previously reported that increased ultrasonication could induce cavitation that might be responsible for a decrease in EE [42]. In a similar way, increased sonication time might disrupt the nano-liposomal structure via the turbulence effect, further leaching MSO out and thus bringing about a lower EE. The 1:8 melon seed oil/Lecithin ratio and 9 min sonication time were implemented for the production of MSO-containing nano-liposomal systems and the encapsulation efficiency was determined as being 57.86 ± 1.76a. The EE of the MSO-containing nano-liposomes was low compared to the EE of chia oil nano-liposomes (88.31 ± 0.64) [15]. The EE of nano-liposomes containing fish oil was determined as being 99.9% [43]. In the context of this study, cholesterol was concurrently used with soy lecithin for the encapsulation of fish oil. The use of an additional coating layer could be responsible for the observed high EE. In another study, the encapsulation of fish oil within a carbohydrate-based wall material resulted in 70% EE [44]. The variations between the observed EEs could be attributed to the difference in methods used for the development of nano-liposomes or the difference in the parameters used in the generation of the nano-capsules and variations in the chemical compositions of the nano-liposomes, such as surfactant usage and wall material to core material ratio. Spray drying and lyophilization are viewed among the common technologies to be used in order to encapsulate the industrially important heat-labile-sensitive compounds. The characteristic properties including size, EE, and aw of nano-liposomal systems and microcapsules are given in Table 6. The EEs of the samples were 74.81 ± 1.14, 64.96 ± 2.72, and 57.86 ± 1.76 for spray-dried samples, lyophilized ones, and nano-liposomal systems, respectively. Size is the main parameter for liposomal systems to be used in biophysical, drug delivery, medicinal, and nutritive studies [45]. Therefore, the sizes of encapsules were measured using a particle sizer. The sizes of liposomes were measured on the nanoscale as expected, while spray-dried and lyophilized samples were found on the microscale. It was reported that nano-vesicles with a diameter above 600 nm could not deliver the targeted material to layers present underneath the skin. Such vesicles tend to remain on the stratum corneum and establish an additional lipid layer on the skin [46,47,48]. Nanocarriers that are smaller than 300 nm in diameter could carry their cargo into the deeper skin layers [49]. Therefore, the nano-liposomes manufactured in this study could be employed to deliver certain nourishing materials to the targeted skin layers besides their use in the enhancement of the nutritional value of foods. The polydispersity index (PDI) represents the magnitude of the heterogeneity of the size distributions within a given mixture. In general, lower PDI values are associated with lower heterogeneity of the nano-liposomal vesicles and PDI values lower than 0.3 are regarded as being acceptable. Diameter size and PDI are the two critical factors highlighting the quality of nano-liposomes. Taken together, our results showed that MSO-carrying nano-liposomes could be regarded as acceptable when considering their size and PDI. In general, a zeta potential in between −30 and +30 mV represents the lower stability of colloidal systems and even instability is elevated with the approach of the value to the zero [50]. The measured zeta potential of the nano-liposomes was −34.70 ± 0.23 mV, which was associated with the enhanced physical stability of the nano-systems. Several investigations have already been published, subjecting oils and essential oils to coating through the use of varied encapsulation methods, as illustrated in Table 7. The thin film hydration method, sonication, ultrasonic homogenization, triazolinedione chemistry, and ethanol injection are among the common techniques that are used to develop nano-sized capsules. The particle size of the oil-carrying capsules ranged from 21.75 nm to 282 nm (Table 7). The encapsulation of linseed oil and linoleic acid resulted in the development of capsules with comparable sizes of MSO-loaded nano-liposomes. The PDIs of the MSO-loaded nano-liposomes were acceptable when compared with the previously reported PDIs which ranged from 0.07 to 0.753 (Table 5), indicating that the thin film hydration method could be used to fabricate nano-vesicles containing MSO. 

### 3.4. Authentication and Morphological Imaging of Nano-liposomal Systems and Microcapsules

FTIR spectroscopy was carried out to further detect the kind of molecules related to the presence of both coating agents and core materials in the capsules. Coating agents (GA-MD-Lecithin), melon seed oil (MSO), nano-liposomal systems, and lyophilized and spray-dried capsules were examined in the solid state. FTIR spectra were recorded for the characterization of capsules containing MSO and coating agents, as shown in Figure 1; the interaction between varied materials was also monitored. The visible peaks representing gum arabic (GA), maltodextrin (MD), and lecithin on the spectrum demonstrated that the chemical structures of these wall materials were not shifted during the development of nano- and microcapsules [26]. A characteristic FTIR spectrum peak located at a wave length of 3298 cm^−1^ was detected in the spectra obtained from GA, MD, and MSOP samples, which is characteristic of hydroxyl (–OH) groups, whereas the mentioned band was not present in the MSO FTIR spectrum and so the presence of that particular band could be postulated as a molecular fingerprint, exhibiting the existence of carbohydrate-based coating agents (GA, MD) in the capsules containing MSO. There were visible functional classes corresponding to the C–H groups, carboxylic group (COOH), C–O, C–C, and C–O–C stretching, and C–O–H, C–H bending on the FTIR spectrum located at 2924.09 cm^−1^, 1639, 1417 cm^−1^, and 759.95–1147.65 cm^−1^, respectively. The visual appearance of the stretching and bending for functional classes on the FTIR spectrum were linked to the polymeric backbone of gum arabic [26,58,59,60,61,62]. There were also stretching vibrations standing for the alkene carbon–hydrogen bond (C–H), CH_2_ carbon–hydrogen bond, carbon–oxygen double bond (–C=O), aromatic rings, and carbon–carbon bonds present in the glucopyranose rings existing in the MD polymers; the corresponding peaks were visible on the FTIR spectrum located at 2924.09 cm^−1^, 2850.82 cm^−1^, 1644.77 cm^−1^, 1417.68 cm^−1^, and 759.95–1147.65 cm^−1^, respectively. While the main IR-band at 2910.87 cm^−1^ observed for gum arabic corresponded to the vibrational modes of the C–H group, the functional class located at 1437 cm^−1^ (GA) might indicate the existence of carboxylic acids located in uronic acid monomers residing on gum polysaccharides [59].

The fingerprinting of MSO was conducted according to the bands only visible on the MSO FTIR spectrum, while they were not detectable on the GA or MD spectra. There were bands corresponding to functional moieties devoted to the triglyceride structure, such as carbon–carbon double bonds (3009.64 cm^−1^) and ester bonds resulting from the condensation of glycerol and fatty acids (1745.58 cm^−1^) [63,64]. The obtained FTIR data indicated the MSO and MD/GA composition present in the microcapsule structure. Similar bands representing the MSO were also apparent in the nano-liposome spectrum. Certain functional classes located in the FTIR spectrum indicated the existence of the lecithin molecule. Stretching vibrations corresponding to visible bands at 3409 cm^−1^ (oxygen–hydrogen bond, O–H) and 1698 cm^−1^ (carbon–oxygen double bond, C=O) were present in the lecithin FTIR spectrum [65]. Additionally, stretching bands equal to the carbon–oxygen bond (C–O, 1170 cm^−1^) and carbon–carbon bond (C–C, 1062 cm^−1^) signalized the lecithin molecule. The existence of IR peaks subtending to lecithin and MSO in nano-liposomal systems also proved the authenticity of the nano-liposomes. The SEM images of microcapsules and nano-liposomal systems are shown in Figure 2. Nano-liposomes and lyophilized microcapsules were freeze-dried prior to SEM imaging, while the spray-dried microcapsules were directly pictured to obtain the SEM images. Most of the spray-dried samples displayed spherical and smooth structures, although there were some spheres with shrunk and irregular shapes. In general, the freeze-dried samples exhibited flakey structures and there were also visible broken lamellar structures in the lyophilized samples. Apart from the spray-dried ones, freeze drying did not bring about the formation of spherical structures. The spraying of the inlet feed before heat application generally results in a fine structure with uniform distribution, as seen in Figure 2 [66].

### 3.5. Thermal Stability of Nano-Liposomal Systems and Microcapsules Containing LCFAs

The thermal stabilities of the nano-liposomal systems and microcapsules were examined using TGA curves. The TGA weight loss curves of MSO, lyophilized capsules, spray-dried capsules, and nano-liposomal systems are given in Figure 3. The deviations in the curves’ shapes propose different mechanisms of degradation among the assayed molecules. The initial mass loss on the TGA graph could be seen as being close to 100 °C, which might be related to a loss in the humidity present in the samples [67]. The TGA results exhibited that the nano-liposomal systems had the highest thermal stability when compared to microcapsules and melon seed oil (MSO); both samples displayed an elevated amount of mass loss in line with the increase in the temperature until 300 °C. The most distinctive decomposition was seen at approximately 300 °C, which is consistent with the thermal degradation profile of complex carbohydrate polymers [68,69]. The rapid degradation continued and 80% of the samples decomposed at around 500 °C. The rapid thermal degradation of MSO started at around 400 °C and almost all of the MSO decomposed close to 500 °C. In the elevated temperature degrees after 500 °C, the nano-liposomal systems displayed the highest stability, followed by the spray-dried and lyophilized samples. Therefore, it can be concluded that nano-liposomal systems display remarkable thermal stability until 300 °C, and it might be claimed that nano-liposomal systems could be incorporated into food matrixes that undergo heat treatment, as 100 °C is a general temperature applied in food sterilization treatments. According to the thermogravimetric analysis results, it was found that nano-liposomal systems harboring MSO displayed outstanding thermal stability. Nano-liposomal systems could easily be used in thermally treated food production processes with no further protection method.

### 3.6. In Vitro Release Characteristics of LFCA-rich MSO-Containing Capsules within the Gastrointestinal Tract

The sustained delivery of linoleic-acid- and oleic-acid-rich MSO and its release characteristics were observed by utilizing an in vitro gastrointestinal system (GIS) simulation. Simulated salivary fluids (SSFs), simulated gastric fluids (SGFs), and simulated intestinal fluid (SIF) phases were established to observe the release characteristics of three different capsule systems. The nano-liposomal system, lyophilized capsules, and spray-dried capsules were separately placed into SSF, SGF, and finally SIF phases, and the release characteristics of the capsules were measured, followed by the measurement of MSO in the fluids. The nano- and microcapsules behaved differently in the SSF, SGF, and SIF phases. The release of LCFA-rich MSO from the spray-dried and lyophilized carbohydrate-coated capsules started in the SSF, while there was no detectable MSO release from the nano-liposomal systems in the SSF phase. These data are compatible with previous results, as carbohydrate digestion begins in the mouth with α-amylase enzymatic activity [13]. The release of LCFA-rich MSO continued in the SGF phase. The highest LCFA-rich MSO release in the SGF phase was observed in the microcapsules manufactured via lyophilization, followed by spray-dried capsules and finally nano-liposomal systems. The residence time of nano-liposomes within the GIS tract could be counted among the main factors, indicating their quality and usability in addition to diameter size, PDI, and zeta potential [70]. The prolonged residence time in the GIS tract is a desired property of nano-carriers to be used in targeted delivery purposes. The release of LFCA-rich MSO from lecithin-based nano-carriers was quite low (5.51%) in the SGF medium, which indicates that nano-liposomal systems slowly release the central core material within the gastrointestinal highly acidic gastric environment. Similar results were reported by Choudhary et al. [15] and only 3.39% of the chia oil was released after 2 h of SGF digestion. It has also been published that nano-liposomes released 13.5 and 12.32% krill oil in the SGF environment [71], which indicates a higher release ratio compared to the results that we obtained in this investigation. At the end of the in vitro digestion system tract, the highest release of LCFA-rich MSO was detected in the carbohydrate-coated microcapsules (29.47–30.35%), while the nano-liposomal systems released only 22% of the total LFCA-rich MSO encapsulated in the liposomes (Table 8). The results obtained in the in vitro release experiments exhibited the fact that nano-liposomal systems are more stable than microcapsules within the GIS tract and exhibit slower release behavior. MSO contains a large amount of LFCAs besides polyphenols, vitamins, and carotenoids. It can be argued that both nano-capsules and microcapsules could be used to deliver MSO in the GI tract to bring these nutrients and non-nutrient molecules into the human body. It has also been reported that nano-sized liposomes (<300 nm) are more viable drug delivery carriers (DDCs), especially in the treatment of cancer cases. Therefore, we have shown that nano-liposomal systems could be developed to deliver nutritive food value compounds and drugs to the GIS tract with a prolonged residence time of LFCAs.

## 4. Conclusions

This investigation describes for the first time the nano-encapsulation and micro-encapsulation of MSO within carbohydrate- and lipid-based wall materials. The thin film dispersion method resulted in the generation of MSO-harboring nano-liposomal systems with acceptable PDI, zeta potential, and low EE. The sonication time and core material to wall material ratio profoundly influenced the EE and it was concluded that 9 min sonication time and 1 to 8 ratios are the best conditions to generate nano-liposomes containing LFCA-rich MSO. The diameter sizes of the nano-liposomes were below 300 nm, which emphasizes their possible use in the delivery of drugs, functional components, and bioactive compounds into the deeper layers. The polydispersity index and zeta potential results indicate homogenous distribution and high stability of the nano-liposomal systems. The thermal stability of the nano-liposomes was higher than that of the microcapsules in the elevated temperature degrees after 450 °C. Based on the results obtained in this study, the encapsulation efficiency and release of MSO in the SGI were low. Further optimization studies are needed to improve the encapsulation efficiency and delivery, as well as in vivo studies to confirm the observations performed in vitro.

## Figures and Tables

**Figure 1 foods-12-02371-f001:**
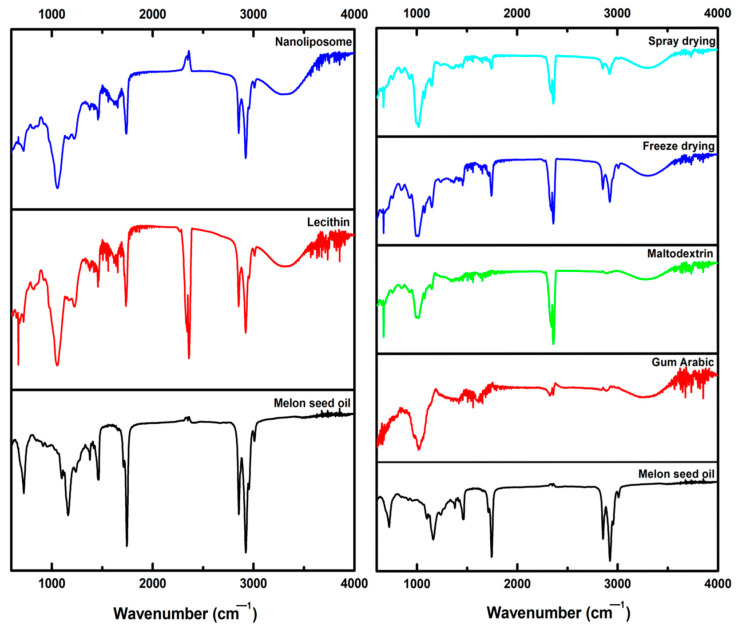
FTIR spectra of nano-liposomal systems and microcapsules.

**Figure 2 foods-12-02371-f002:**
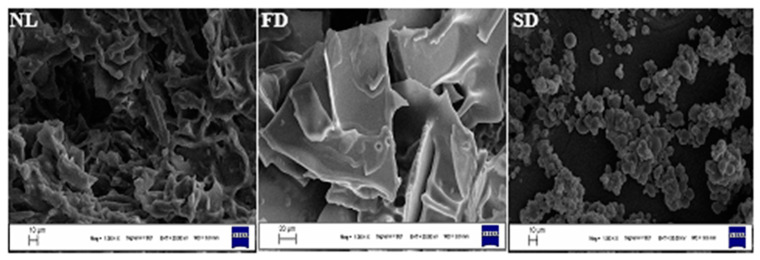
SEM images of encapsules (NL: nano-liposomes, FD: freeze-dried samples, SD: spray-dried samples).

**Figure 3 foods-12-02371-f003:**
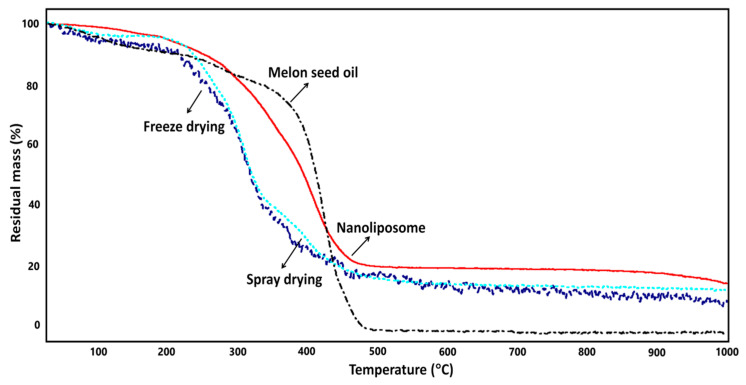
TGA curves of melon seed oil and encapsulated systems.

**Table 1 foods-12-02371-t001:** Oil yield and oxidation values of melon seed oils obtained using different extraction methods.

Methods	Oil Yield (%)	Peroxide Value (meq/kg oil)	*p*-Anisidine	TOTOX Value	K_232_	K_270_
SE	18.75 ± 0.50 ^a^	11.96 ± 0.86 ^c^	4.67 ± 0.05 ^c^	28.59 ± 1.12 ^c^	0.99 ± 0.05 ^d^	0.93 ± 0.02 ^d^
MWAE	18.89 ± 0.15 ^a^	5.01 ± 0.10 ^b^	3.02 ± 0.01 ^b^	13.05 ± 0.21 ^b^	0.77 ± 0.04 ^c^	0.56 ± 0.07 ^c^
USAE	22.85 ± 0.11 ^b^	3.96 ± 0.03 ^a^	2.19 ± 0.12 ^a^	10.11 ± 0.22 ^a^	0.45 ± 0.01 ^a^	0.11 ± 0.04 ^a^
UMAE	19.11 ± 0.10 ^a^	4.67 ± 0.19 ^ab^	2.79 ± 0.16 ^b^	12.14 ± 0.55 ^b^	0.69 ± 0.03 ^b^	0.44 ± 0.02 ^b^

SE: Soxhlet extraction, MWAE: microwave-assisted extraction, USAE: ultrasound-assisted extraction, UMAE: combination of microwave- and ultrasound-assisted extraction. The lower case letters in the same column exhibit differences among the treatments.

**Table 2 foods-12-02371-t002:** The effect of extraction methods on the color parameters of melon seed oils.

Methods	L*	a	b
SE	34.84 ± 0.03 ^c^	0.12 ± 0.07	9.24 ± 0.10 ^a^
MWAE	32.80 ± 0.06 ^a^	0.17 ± 0.04	10.69 ± 0.10 ^c^
USAE	33.04 ± 0.04 ^ab^	0.22 ± 0.07	10.49 ± 0.02 ^b^
UMAE	33.53 ± 0.60 ^b^	0.29 ± 0.12	10.39 ± 0.07 ^b^

SE: Soxhlet extraction, MWAE: microwave-assisted extraction, USAE: ultrasound-assisted extraction, UMAE: combination of microwave- and ultrasound-assisted extraction. The lower case letters in the same column exhibit differences among the treatments.

**Table 3 foods-12-02371-t003:** Bioactive properties of melon seed oils obtained using different extraction methods.

Methods	Total Phenolic Content (mg GAE/100 g)	DPPH (µmol TE/g)	ABTS (µmol TE/g)	FRAP (µmol TE/g)	CUPRAC (µmol TE/g)
SE	29.71 ± 1.32 ^a^	0.37 ± 0.40 ^a^	4.98 ± 0.18 ^a^	0.99 ± 0.02 ^a^	4.94 ± 0.11 ^a^
MWAE	50.01 ± 3.68 ^b^	0.95 ± 0.15 ^b^	9.54 ± 0.03 ^b^	2.89 ± 0.16 ^b^	8.36 ± 1.16 ^b^
USAE	70.14 ± 0.53 ^d^	1.72 ± 1.88 ^c^	10.93 ± 0.06 ^c^	4.49 ± 0.06 ^d^	29.10 ± 2.66 ^d^
UMAE	54.30 ± 1.84 ^c^	1.19 ± 0.10 ^b^	10.11 ± 1.14 ^c^	3.14 ± 0.02 ^c^	15.02 ± 0.08 ^c^

SE: Soxhlet extraction, MWAE: microwave-assisted extraction, USAE: ultrasound-assisted extraction, UMAE: combination of microwave- and ultrasound-assisted extraction. The lower case letters in the same column exhibit differences among the treatments.

**Table 4 foods-12-02371-t004:** Fatty acid compositions of melon seed oils obtained using different extraction methods.

Fatty Acids (%)	SE	MWAE	USAE	UMAE
Capric acid	0.02 ± 0.0	0.01 ± 0.0	0.02 ± 0.0	0.02 ± 0.0
Tridecanoic acid	nd	nd	nd	nd
Myristic acid	0.06 ± 0.0	0.05 ± 0.0	0.05 ± 0.0	0.07 ± 0.0
Palmitic acid	9.61 ± 0.3	9.95 ± 0.4	9.70 ± 0.3	9.63 ± 0.3
Palmitoleic acid	nd	nd	nd	nd
Stearic acid	1.96 ± 0.05	1.97 ± 0.06	1.97 ± 0.04	1.99 ± 0.07
Cis-oleic acid	15.24 ± 0.6	14.98 ± 0.5	15.21 ± 0.2	15.25 ± 0.4
Cis-linoleic acid	73.10 ± 1.1	73.95 ± 1.2	73.56 ± 0.9	73.11 ± 1.2
Linolenic acid	nd	nd	nd	nd
Cis-4,7,10,13,16,19-docosahexaenoic acid	nd	nd	nd	nd
Total saturated fatty acids	11.65	11.98	11.74	11.71
Total mono-unsaturated fatty acids	15.24	14.98	15.21	15.25
Total polyunsaturated fatty acids	73.10	73.95	73.56	73.11
Total unsaturated fatty acids	88.34	88.93	88.76	88.36
SFA/USFA	0.13	0.14	0.13	0.13

SE: Soxhlet extraction, MWAE: microwave-assisted extraction, USAE: ultrasound-assisted extraction, UMAE: combination of microwave- and ultrasound-assisted extraction, nd: not detected.

**Table 5 foods-12-02371-t005:** Experimental conditions and responses for nano-liposomal systems.

Melon Seed Oil/ Lecithin Ratio	Sonication Time (min)	Encapsulation Efficiency (%)
1:1	6	41.18 ± 0.95 ^d^
1:4	6	45.36 ± 0.71 ^c^
1:8	6	51.97 ± 1.02 ^b^
1:16	6	52.03 ± 0.98 ^b^
1:8	1	28.12 ± 1.12 ^f^
1:8	3	34.46 ± 1.46 ^e^
1:8	6	52.24 ± 1.14 ^b^
1:8	9	57.86 ± 1.76 ^a^
1:8	12	55.82 ± 1.92 ^a^

Different letters in the same column indicate statistically significant differences (*p* < 0.05).

**Table 6 foods-12-02371-t006:** Characteristic properties of samples.

Properties	Nano-liposome	Freeze Drying	Spray Drying
Particle Size (nm)	282.30 ± 2.35	3140 ± 12	2660 ± 14
Polydispersity Index	0.249 ± 0.011	-	-
Zeta Potential (mV)	−34.70 ± 0.23	-	-
Encapsulation Efficiency	57.86 ± 1.76	64.96 ± 2.72	74.81 ± 1.14
Water Activity (a_w_)	0.390 ± 0.001	0.405 ± 0.001	0.326 ± 0.001

**Table 7 foods-12-02371-t007:** Preparation method, particle size, and polydispersity index of encapsulated oils and essential oils.

Oil Type	Technique	Particle Size (nm)	PDI	Reference
Melon Seed Oil	Thin Film Hydration and Sonication	282.30 ± 2.35	0.249 ± 0.011	[This study]
Chia Oil	Thin Film Hydration and Sonication	49.25	0.175	[15]
Orange Essential Oil	Ultrasonic Homogenization	21.75	0.753	[51]
Garlic Essential Oil	Thin Film Hydration and Sonication	101	0.127	[52]
Almond Oil	Spray Drying	1.4–31.1	-	[53]
Olive	Triazolinedione Chemistry	185	0.11	[54]
Pumpkin Oil	Triazolinedione Chemistry	172	0.08	[54]
Sunflower Oil	Triazolinedione Chemistry	178	0.07	[54]
Hazelnut Oil	Triazolinedione Chemistry	175	0.09	[54]
Cardamon	Thin Hydration Method	71.8–147.9	>0.261	[25]
Coix Seed Oil + β-carotene	Ethanol Injection	156–193	<0.30	[55]
Linoleic Acid	Ethanol Injection	266	<0.30	[56]
Linseed Oil + Quercetin	Ethanol Injection	262	-	[57]

**Table 8 foods-12-02371-t008:** The release ratio of essential oil in simulated gastrointestinal fluid.

Conditions	Nano-Liposome	Freeze Drying	Spray Drying
SSF (%)	-	0.92 ± 0.01	1.34 ± 0.01
SGF (%)	5.51 ± 0.05	8.82 ± 0.05	9.07 ± 0.05
SIF (%)	22.12 ± 0.10	29.47 ± 0.10	30.35 ± 0.12

SSF: simulated saliva fluid, SGF: simulated gastric fluid, SIF: simulated intestinal fluid.

## Data Availability

The data used to support the findings of this study can be made available by the corresponding author upon request.

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
