# Peer review of "Nano- and Micro-Encapsulation of Long-Chain-Fatty-Acid-Rich Melon Seed Oil and Its Release Attributes under In Vitro Digestion Model"

_foods, 2023, doi:10.3390/foods12122371_

Round 1

Reviewer 1 Report

Dear authors: I found your manuscript very interesting. I think that is well written and developed. A complete comparation of micro and nano encapsulation of melon seed oil was correctly performed. The experimental development was very complete exploring several methodologies of characterization. In order to improve the quality of the mansucript, some changes are suggested in pdf file.

Author Response

Dear Reviewer,

I would like to thank you for the valuable comments you made on the manuscript entitled with ‘Constructing Long Chain Fatty Acid Harboring Nanoliposomal Delivery Systems and Investigation of Release Attributes within In Vitro Digestion Model’ which was submitted to FOODS journal for publication. Your comments helped me to improve the quality of the paper. I have made the following changes to meet the raised points and hope that you find the manuscript worth to publish with these changes. 

Reviewer 2 Report

This paper attempted to compare the nutritional quality of melon seed oil extracted using 4 different extraction methods and study the stability of two different encapsulation methods. Generally this paper failed to discuss the importance of this study and results cannot be confirmed because no significant differences between compared groups were included. Thus, discussions and conclusions are irrelevant.

Title: Does not depict the extraction methods and the main material - melon seeds.

Introduction: Description of economical values and sustainability in using fruit by-products should be written in the first paragraph. Then the authors should explain why melon seeds (from what species?) was chosen as the starting material compared to other fruit by-products.

Methodology: There are no reference to indicate the reason for choosing the temperature 80 celcius and 30 minutes time of extraction for the treatments.

Results and Discussion: Cannot be verified because of lack of report for statistical significance in the tables provided.

Author Response

(The authors gave the same response as above.)

Reviewer 3 Report

The idea of the manuscript is overall interesting. Here are some suggestions for improvement:

- Title: I think the title should be reformulated. I believe that the authors could consider to include the word "melon seed oil" in the title to have a more representative title with regard to their study.

- Part 2.4: please describe the methodology in detail even though briefly, not just put citations

- Table 1 is not necessary. The authors could just incorporate the information in the text

- Table 3: please remove the asterisk signs (*) from a and b. 

Author Response

(The authors gave the same response as above.)

Reviewer 4 Report

Dear Editor nad Authors,

The manuscript ‘Constructing Long Chain Fatty Acid Harboring Nanoliposomal Delivery Systems and Investigation of Release Attributes within In Vitro Digestion Model’ by Asliye Karaaslan describes research on encapsulation of melon seed oil. The manuscript is well written, methods appropriate and the topic is interesting for the readers.

However,  I recommend minor revision.

Line 12 encapsulation is a technology to provide thermal (…) fro functional…

Line 41 Latin names in italics

Line 85 Use past simple: encapsulation techniques were engaged

Line as above: the capluses containing MSO were measured

Line 99 Soxlet

Line 100 Give producers of rotary evaporator and vacuum drying oven

Tables No dot should be used in Tables’ titles at the end. Table’s title is opening the table.

No statistics is marked in the Tables 1-5 and 7. Are all the data equal, the same? The comments suggest that no they are not. Please add the statistics.

Yours sincerely,

Dear Editor and Authors,

In my opinion a moderate editing of English would be useful. Most comments I have attached above,

Yours sincerely,

Author Response

(The authors gave the same response as above.)

Reviewer 5 Report

The paper fits the journal; however, there is a major lack, the comparison with previous research studies. Please include a table where you compare the main findings of this project with previous ones.

Need to be improved.

Author Response

(The authors gave the same response as above.)

Round 2

Reviewer 1 Report

The suggestions and corrections were taken into account by the authors and the text was modified accordingly. I think the quality of the manuscript has improved

Author Response

The author would like to thank the reviewer for the comment.
